# Non-Native Decapods in South America: Risk Assessment and Potential Impacts

Lucas Rieger de Oliveira [1,*], Gustavo Brito [2], Mafalda Gama [3], Ximena María Constanza Ovando [1], Pedro Anastácio [3] and Simone Jaqueline Cardoso [1,4,*]

1 Graduate Program in Biodiversity and Nature Conservation, Institute of Biology, Federal University of Juiz de Fora, Juiz de Fora 36036-900, MG, Brazil; velliger08@gmail.com

2 Graduate Program in Biosciences, Aquatic Biology Laboratory (LABIA), São Paulo State University, Assis 19806-900, SP, Brazil; reis.brito@unesp.br

3 Department of Landscape, Environment and Planning Marine and Environmental Sciences Centre (MARE)/Aquatic Research Network (ARNET), Institute for Research and Advanced Training (IIFA), University of Évora, 7002-554 Évora, Portugal; mafaldagama@hotmail.com (M.G.); anast@uevora.pt (P.A.)

4 Department of Zoology, Institute of Biology, Federal University of Juiz de Fora, Juiz de Fora 36036-900, MG, Brazil

* Correspondence: lucas.rieger2014@gmail.com (L.R.d.O.); simone.jcardoso@gmail.com (S.J.C.)

**Abstract:** Biological invasions pose significant challenges in the Anthropocene, impacting ecosystem biodiversity and functioning. Ecological Niche Modeling is widely used to evaluate potential areas at risk of invasions, aiding in the prevention of invasive-species expansion and guiding conservation efforts in freshwater ecosystems. The main objectives of this study were to model the ecological niche and evaluate remaining suitable habitat areas for the occurrence of five potentially invasive species of freshwater decapods in South America: *Dilocarcinus pagei*, *Macrobrachium amazonicum*, *M. jelskii*, *M. rosenbergii*, and *Procambarus clarkii*. Occurrence data from the Global Biodiversity Information Facility were complemented with a literature systematic review. Variables used in the models were obtained from the Worldclim and EarthEv databases. Ecological Niche Modeling was performed using the *Biomod2* and *sdm* package algorithms. Our results indicated a suitable area of up to 11% of South America. Model evaluations yielded favorable TSS and AUC values (>0.7 and >0.8). The suitable areas projected for South America included several hydrographic basins and Protected Areas. The information generated in our study can help identifying areas susceptible to decapod invasion in South America and support local management and decisions.

**Keywords:** biological invasions; crustaceans; niche modelling

## 1. Introduction

The introduction of species into environments outside their natural distributions has caused significant ecological damage in several ecosystems, representing the second leading cause of biodiversity loss worldwide [1,2]. The presence of non-native species negatively affects native counterparts, causes extinctions, and disrupts the biotic structure of the ecosystem, while also affecting the economies of several countries [3,4].

In South America, the number of non-native and invasive freshwater species has increased in the last decade, mainly due to climate change and the degradation of aquatic ecosystems such as predatory fishing and urbanization [5–8]. Non-native species can tolerate a wide range of environmental conditions due to their ecological plasticity, having advantages in resource competition with native fauna, drastically reducing local populations, and leading to significant biodiversity loss and extinction [9].

Freshwater macro- and microcrustaceans have a high invasive potential and are linked to the exploitation of natural resources and anthropogenic impacts, in particular decapods, which are frequently associated with aquaculture in developing countries [10,11]. Decapods

are adapted to freshwater ecosystems, tolerating different salinities and temperature variations which gives them an advantage in colonizing new environments by affecting their reproductive cycle and embryonic and larval development [12–14]. Additionally, several non-native species share the characteristic of tolerating extreme or atypical conditions during their reproductive cycles [15,16].

The South American subcontinent covers more than 17 million km$^2$ and represents approximately 23% of all freshwater bodies in the world, making it one of the world's most biodiverse regions [17]. Furthermore, the introduction of non-native species and the possibility of biological invasions are constant threats to South America's biodiversity. Due to its large territorial extension and latitudinal amplitude, South America has different climates, ranging from intense humidity in the Amazon Forest to strong aridity in the Atacama Desert and the Caatinga. It also presents an extensive combination of terrains that interfere with terrestrial and aquatic ecosystems. Thus, the selection of climate, topographic and hydrological variables is important to evaluate the possible impacts of non-native decapod species on South American aquatic ecosystems [18,19].

In South America, *P. clarkii* (Girard 1852), *Macrobrachium amazonicum* (Heller 1861), *M. jelskii* (Miers 1878), *M. rosenbergii* (De Man 1879), and *D. pagei* (Stimpson 1861), are five important freshwater decapods with a strong invasive potential. *Procambarus clarkii* and *M. rosenbergii* are native to North America and Asia, respectively, and were introduced in South America for commercial purposes, while *M. amazonicum*, *M. jelskii* and *D. pagei* are native to the Amazon region, but can be considered transplants in other regions of South America, because their introductions were provided by anthropogenic actions [20].

The non-native species, *P. clarkii* and *M. rosenbergii*, are widely distributed globally, and their impacts on the native species' community are well-known [21–24]. The occurrence of these species in protected areas can have long-term negative effects on aquatic ecosystems, reinforcing the need for careful attention to the cultivation of this species. *Procambarus clarkii* is considered one of the most invasive crustacean species in the world, and its introduction outside its natural environment has mainly been for commercial purposes in aquaculture or as a pet [25]. Thus, *P. clarkii* has caused impacts on various sectors of the economy, such as rice fields, damaging fishing nets, and negatively affecting amphibian communities and aquatic plant diversity [26,27]. Additionally, *P. clarkii* can be a vector of diseases and an important transmitter of the crayfish plague, which is of great concern to the native crayfish community [28,29].

The other species, *M. amazonicum*, *M. jelskii*, and *D. pagei,* are native to the Amazon region [26,30]. The species *Macrobrachium jelskii* and *M. amazonicum* were introduced to serve as food for some species of fish previously introduced in culture tanks and reservoirs in the region [31], while the introduction of *D. pagei* in other regions of South America is maybe accidental, through the transport of fish and macrophytes [26], and there are records of these species in some basins in the southeast region of Brazil and close to the Pantanal biome. Most of the research carried out in South America on non-native decapods is limited to studies on shrimp farming, and little is known about the potential of farmed species to invade inland waters and cause negative impacts [32–34].

Currently, the use of Ecological Niche Modeling (ENM) tools provides important knowledge for monitoring and controlling the establishment of non-native species, as well as supporting decision making on habitat conservation and priority areas for preservation [35–37]. ENMs are increasingly used in geoprocessing and biogeography projects for mapping areas with a high risk of biological invasions [38,39]. ENM associates biogeographic and climate data with species distribution to predict suitable areas for species occurrence [40,41].

MaxEnt is highly popular in ENM studies for its predictive capacity, bias reduction, and use of background points to make accurate predictions even with incomplete information, making it suitable for modeling large projection areas [42]. Recent studies have used MaxEnt to review species pseudo-absences and found it to be more robust and effective than other methods [43,44].

Therefore, the aims of this study were to (i) model the ecological niches of three transplant species of freshwater decapods in South America: *D. pagei, M. amazonicum, M. jelskii* and two non-native species *M. rosenbergii and P. clarkii*; and (ii) determine the extent of suitable habitat areas for the occurrence of those species in South America.

## 2. Materials and Methods

### 2.1. Study Area

The South America subcontinent territory is located at tropical latitudes, under the influence of the Intertropical Convergence Zone and related circulation systems, creating diverse patterns of weather, climate and climate variability [45]. Three large freshwater basins dominate the South American subcontinent: Amazon, Orinoco and la Plata, and four structural elements are relevant to the shape and behavior of these three large basins: (1) the Andes; (2) foreland basins in the eastern Andes south of the Orinoco to the Chaco-Paraná basin; (3) the Guiana and Brazilian shields reflecting Precambrian cratons and orogenic belts of metamorphic rocks; and (4) the Central Amazon Basin, a large cratonic descendant with some graben structures dating to the early Paleozoic era, connecting the foreland basins to the west with a graben that locates the Amazon estuary on the Atlantic coast [46].

### 2.2. Species Selection

The study focuses on five freshwater decapod species with high potential for invasion in South America, including the already invasive *P. clarkii* and *M. jelskii,* which cause negative impacts such as predation and competition with native species and *D. pagei, M. rosenbergii* and *M. amazonicum*, classified as transplant species with great potential for invasion [5,26,47–52].

*Dilocarcinus pagei* is a freshwater crab with a high osmoregulation capacity and resistance to different environments, potentiating the possibility of it becoming an invasive species [53]. Although native to the Amazon region, as is also true for *M. jelskii* and *M. amazonicum,* there are no records of this species being cultivated for economic and commercial purposes. The apparent lack of commercial interest may decrease the likelihood of deliberate introductions in new regions. However, accidental introductions through the transport of fish or macrophytes could explain the presence of this species in the Paraná River basins, Pantanal, and other regions in South America [26,50].

*Macrobrachium amazonicum* is a species native to the Amazon region, with its natural distribution in the hydrographic basin of the Amazon and Orinoco River [54,55]. This species is a notable representative within its genus because it can inhabit inland areas and is not limited to coastal and estuarine regions [51]. It is considered the shrimp species with the greatest potential for commercial cultivation in the region, which increases the possibility of being introduced outside of its native area [56].

*Macrobrachium jelskii* has habitat and morphological characteristics very similar to those of *M. amazonicum*. However, its distribution is more restricted to coastal areas from Venezuela to the Bahia and Espírito Santo states in Brazil. It does not adapt as easily to inland waters as *M. amazonicum*, although there are occasional records of this species in the interior of the subcontinent, such as in Brazil (Minas Gerais and São Paulo States) and Bolivia [26,57].

*Macrobrachium rosenbergii* is a freshwater shrimp native to inland waters of the Indo-Pacific region and widely distributed in India, Vietnam, and Indonesia [58]. It was introduced in South America in the mid-1980s in the Amazon region, and in recent years, concerns have been raised about its potential impacts [59,60]. There are occasional records of this species in northeastern Brazil (Gurupi and Parnaíba Rivers) and the Paraná River basin near to Argentina's border [61–63].

*Procambarus clarkii* is a crayfish native to North America, widely distributed in northern Mexico and the south-central United States [64] with few records in South America; its presence in Colombia and in São Paulo state (Brazil) may signal its invasive potential [65,66].

## 2.3. Species Occurrence Data

For the systematic review and to compile data of the presence of five species, four databases were explored—Web of Science www.webofscience.com (accessed on 29 January 2021), Scopus www.scopus.com (accessed on 29 January 2021) Scielo https://www.scielo.br (accessed on 29 January 2021), and Aquatic Science and Fisheries Abstract (ASFA), proquest.libguides.com/asfa (accessed on 29 January 2021). The same search term was used across all databases: "(*Procambarus* OR *Macrobrachium* OR *Dilocarcinus*) AND (estuar* OR wetland* OR mangrove* OR freshwater OR "aquatic environment" OR "aquatic ecosystem*" OR lake* OR river* OR pond* OR reservoir* OR "drainage basin*" OR lagoon* OR "river basin*" OR stream* OR waterfall* OR watercourse* OR brook* OR creek*)". The systematic review used peer-reviewed articles published from 1945 to January 29th, 2021. All articles found in the four databases search were exported to a .RIS file, classified, and duplicates were removed using EndNote Program (version x9.3.2) [67]. These articles were exported to Microsoft Excel [68] to apply the inclusion and exclusion criteria. All articles with species occurrence records with geographic coordinates were included. Articles without geographic coordinates, but with location descriptions, were separated and later georeferenced using the Google Earth website earth.google.com/2022 (accessed on 1 July 2022). Experimental studies with captive animals, laboratory or aquaculture tanks, reports, scientific notes (except new species records), pre-prints, and book chapters were not included in the screening. The remaining occurrences were collected from the records published in the Global Biodiversity Information Facility (GBIF 2022) database. A PRISMA matrix [69] was generated to provide a better visualization of the results and to detail the occurrence data collection. The main records were those found through the systematic review of the literature, while additional records were obtained from GBIF occurrences.

## 2.4. Niche Modeling and Variable Selection

The occurrence data of selected species were statistically combined with biogeographical predictor variables extracted from WorldClim worldclim.org (accessed on 1 July 2022) and EarthEnv earthenvi.org (accessed on 1 July 2022) databases to predict the species' suitable niches based on the selected environment [70]. As EarthEnv variables are only available at a resolution of 30 arc-second (1 km) and Worldclim variables at a resolution of 2.5 arc-minute (4.5 km), it was necessary to standardize all predictor variables to approximately 5 km$^2$ for all layers using the raster package [71] on R-program (4.1.2 version) [72] and the QGIS program (3.16.16 version) [73]. The variables were cropped with consideration to the South America shapefile as a "mask" using the tidyverse [74] and rnaturalearth [75] packages.

The EarthEnv data is available in high resolution for rivers and lakes and shows monthly average values over each water body [76,77]. However, for large lakes where conditions can be highly variable, the raster presents a single value that does not adequately cover the range of environmental conditions and habitats available for freshwater environments. To address this issue, the 19 Worldclim bioclimatic variables were also added in the modeling. This approach has been previously used in modeling studies for various non-native or invasive species [38,78,79].

The EarthEnv variables were selected based on the species' biology, therefore hydroclimate, topography, flow, and precipitation were used. All variable layers were extracted from netCDF-4 files available on the EarthEnv platform and were clipped to the same extension and resolution as the Bioclimatic variables. Once standardized, the variables were included in the same dataset, totaling 60 predictor variables from both databases. After including these layers into a single dataset, multicollinearity between the variables was assessed using the Variance Inflation Factor (VIF) in the usdm package [80] of the R-project. Finally, highly correlated variables were excluded from the niche modeling, with the default cutoff set at greater than or equal to 10 to avoid collinearity in statistical models [80].

### 2.5. Modeling Protocol

All occurrence records of the selected species were filtered to ensure unique coordinates and exclude problematic points, such as those in oceans, biased towards large institutions, capitals, redundant data, and duplicates. To further remove any sampling and spatial biases that could negatively impact model construction, the spThin package was used [81]. To perform ENM for the five decapods species, predictive variables for the entire globe were selected in the same way as the occurrence records.

The niche modelling for all species was carried out based on their respective native regions and projected onto the area of interest. For example, to perform niche modeling of *P. clarkii*, the model was built using North America and projected onto South America. Likewise, the niche modelling for *M. rosenbergii* was performed for the entire Asian continent and projected onto South America. This approach aims to minimize spatial sampling bias and avoid the overfitting of the models, which can occur when using species records from the entire globe. In this sense, a more reliable and realistic model is expected.

Models were evaluated using the area under the ROC (receiver operating characteristic) curve (AUC) and the true skill statistics (TSS). AUC values range from 0 to 1, with values of 0.9–1 considered excellent and 0.8–0.9 considered good. TSS values are threshold-dependent and can range from −1 to +1, with positive values ranging from 0.2 to 0.5 considered poor, from 0.6 to 0.8 considered useful, and values greater than 0.8 considered excellent [82,83].

The random selection of pseudo-absence points was used [84] as it was observed that these algorithms tend to perform better because they manage to reduce spatial bias and select points where the species have low suitability for occurrence [44]. In this sense, the number of pseudo-absences was standardized for each species to reduce this bias further, since an insufficient or excessive number of pseudo-absences can generate over-prediction models, which produce maps with either very high suitability for the species or with no biological sense. Thus, the models were run using 10 replicates of each randomly generated pseudo-absences set with the default number of 1000 pseudo-absences for all species. Records were entered using a 2.5 arc-minutes resolution for South America and randomly divided into test and training, using 20% and 80% of the data, respectively, according to Thuiller's proposal [85].

The nine algorithms selected from the Biomod2 package were generalized linear models (GLM) [86], generalized boosted models (GBM) [87], generalized additive models (GAM) [86], classification tree analysis (CTA) [88], artificial neural networks (ANN) [89]), surface range envelope (SRE) [90], flexible discriminant analysis (FDA) [91], multivariate adaptive regression spline (MARS) [92], and random forest (RF) [93]. These algorithms generated predictions using the default settings of the Biomod2 model [94]. Individual predictions were then averaged to create an ensemble approach, with the four algorithms showing the best performance selected based on the average TSS and AUC values for all species. The four highest means of the TSS individual algorithm, with a cutoff value of 0.7, and the four highest means of the AUC individual algorithm, were selected to produce the final ensemble approach [95].

In addition, we produced a second model using the MaxEnt algorithm that used background data as absence data. This model was created using the sdm package [95] and default settings of 10,000 background points [43]. The modeling was performed at a resolution of 2.5 arc minutes and divided randomly into test and training subsets, with 20% and 80%, respectively, similar to the Biomod2 package. We identified the contribution of each variable to the model for each species using the default settings of the Biomod2 and sdm packages, which selects variables using Pearson's correlation. We generated suitability maps by combining the means of each generated model using the four best-performing algorithms according to the TSS value for the ensemble [95]. AUC values were also considered for algorithm selection, and only algorithms with an AUC greater than 0.8 were selected for the ensemble. The models used were classified as good or excellent [96,97]. The dataset of the models was grouped into a single map through the raster and terra

packages [71] in R. For each species, we calculated the total suitability area using the field calculator and the GRASS GIS r.class tools [98] in QGIS, with a cut-off point above 0.75. Finally, priority areas with a high risk of invasion were defined by superimposing the suitability areas of each species on the Protected Areas (PAs) and Watersheds shapefile layers, available in the databases of the International Union for Conservation of Nature (IUCN 2021).

## 3. Results

### 3.1. Species Presence Data

The systematic review of the literature resulted in 10,850 articles from the four databases, obeying all inclusion criteria and search terms. After removing the duplicates using Endnote, 9597 unique articles that were exported to Microsoft Excel (2306 version, 2016). Further screening excluded 8592 articles that did not meet the inclusion criteria, leaving 1005 articles that contained species locations. After the final screening step, which removed duplicate coordinates and papers with identical locations, there were 551 articles remaining. All articles selected in the data screening are included in Supplementary Table S1.

The systematic review data are presented in the PRISMA flow diagram in Supplementary Figure S1. Through the literature review, we found approximately 20% more occurrence records compared to those obtained through GBIF ($n = 6801$). In total, we recorded a total of 1367 occurrences which were included in the occurrence-record spreadsheet. Out of these records, there were 42 unique coordinates for *D. pagei*, 213 for *M. amazonicum*, 116 for *M. jelskii*, 105 for *M. rosenbergii*, and 908 for *P. clarkii*. The considerable increase in records of occurrences, obtained through the systematic review of the literature, can be observed in the maps of Supplementary Figure S2, resulting in the better performance of the models as the number of occurrences increased.

### 3.2. Niche Modeling

All the variables selected (bioclimatic, hydrological, topographic, flow, and precipitation) by VIF to create the ENM for each species are listed in the table provided in Supplementary Table S2. Despite the exclusion of many predictor variables, the only common variable among all five species after VIF selection was the sum of monthly precipitation in November (prec11). The other variables that were present in at least four of the five species were the minimum elevation (Elev1), the precipitation of the warmest quarter (BIO18), and the average diurnal interval for hydrological variables (hydro2) (Table 1).

**Table 1.** Average of variables' importance for the construction of the models of each species calculated by Pearson's correlation. *P. clarkii = Procambarus clarkii, D. pagei = Dilocarcinus pagei, M. rosenbergii = Macrobrachium rosenbergii, M. jelskii = Macrobrachium jelskii, M. amazonicum = Macrobrachium amazonicum.* bio1 = Annual Mean Temperature, bio4 = Temperature Seasonality, bio8 = Mean Temperature of Wettest Quarter, bio9 = Mean Temperature of Driest Quarter, bio13 = Precipitation of Wettest Month, bio14 = Precipitation of Driest Month, bio18 = Precipitation of Warmest Quarter, prec2 = Sum of monthly precipitation February, prec5 = Sum of monthly precipitation May, prec7 = Sum of monthly precipitation July, hydro2 = Mean Diurnal Range, hydro9 = Mean Temperature of Driest Quarter, hydro14 = Precipitation of Driest Month.

| Species | Variable Importance | | |
|---|---|---|---|
| *M. amazonicum* | hydro14 (0.26) | bio4 (0.25) | hydro2 (0.18) |
| *D. pagei* | bio14 (0.41) | bio8 (0.39) | bio18 (0.27) |
| *M. jelskii* | prec5 (0.55) | hydro2 (0.49) | bio9 (0.45) |
| *P. clarkii* | bio1 (0.49) | prec7 (0.12) | prec2 (0.07) |
| *M. rosenbergii* | bio13 (0.31) | hydro9 (0.24) | bio8 (0.2) |

The temperature and precipitation variables had a significant influence on the most important predictor variables for constructing the models for each species (Table 1), with

the terrestrial variables of WorldClim being important for all species. Additionally, precipitation, elevation, and hydrology were important for at least one species. We also noted the November precipitation variable (prec11) which was selected by VIF for all species in the present study (Table S2) but was not considered as one of the three most important variables (Table 1).

Table 2 displays the mean TSS and AUC values for each algorithm used in the modeling performed with the Biomod2 [84] and sdm [95] packages. The algorithms with the highest mean TSS and AUC values were RF (0.86) and (0.96), GBM (0.82) and (0.95), GAM (0.77) and (0.92), and GLM (0.76) and (0.90), respectively. The other algorithms (SRE, CTA, MARS, FDA, and ANN) that were considered less predictive by the model evaluation were not included in the ensemble.

**Table 2.** Mean AUC and TSS for each algorithm used in the niche modeling of the species *M. rosenbergii*, *P. clarkii*, *M. jelskii*, *D. pagei*, and *M. amazonicum*. GLM = generalized linear models, GBM = Generalized Boosted Models, GAM = generalized additive models, CTA = classification tree analysis, ANN = artificial neural networks, SER = surface range envelope, FDA = flexible discriminant analysis, MARS = multivariate adaptive regression spline, RF = random forest. *P. clarkii* = *Procambarus clarkii*, *D. pagei* = *Dilocarcinus pagei*, *M. rosenbergii* = *Macrobrachium rosenbergii*, *M. jelskii* = *Macrobrachium jelskii*, *M. amazonicum* = *Macrobrachium amazonicum*.

| Algorithm | *D. pagei* | | *M. amazonicum* | | *M. jelskii* | | *M. rosenbergii* | | *P. clarkii* | |
|---|---|---|---|---|---|---|---|---|---|---|
| | TSS | ROC | TSS | ROC | TSS | ROC | TSS | ROC | TSS | ROC |
| SRE | 0.52 | 0.76 | 0.42 | 0.71 | 0.27 | 0.64 | 0.44 | 0.72 | 0.51 | 0.76 |
| CTA | 0.60 | 0.78 | 0.60 | 0.71 | 0.65 | 0.84 | 0.80 | 0.91 | 0.83 | 0.94 |
| **RF** | **0.85** | **0.94** | **0.82** | **0.95** | **0.83** | **0.95** | **0.89** | **0.97** | **0.89** | **0.98** |
| MARS | 0.76 | 0.90 | 0.73 | 0.92 | 0.71 | 0.90 | 0.81 | 0.94 | 0.80 | 0.96 |
| FDA | 0.64 | 0.84 | 0.68 | 0.89 | 0.69 | 0.89 | 0.81 | 0.92 | 0.80 | 0.96 |
| **GLM** | **0.76** | **0.84** | **0.71** | **0.91** | **0.76** | **0.85** | **0.79** | **0.93** | **0.81** | **0.95** |
| **GBM** | **0.83** | **0.95** | **0.79** | **0.94** | **0.78** | **0.93** | **0.87** | **0.97** | **0.85** | **0.97** |
| GAM | **0.71** | **0.90** | **0.77** | **0.92** | **0.77** | **0.90** | **0.77** | **0.89** | **0.84** | **0.97** |
| ANN | 0.72 | 0.88 | 0.53 | 0.80 | 0.54 | 0.79 | 0.70 | 0.86 | 0.74 | 0.89 |
| MaxEnt ** | 0.73 | 0.84 | 0.82 | 0.94 | 0.72 | 0.94 | 0.77 | 0.93 | 0.75 | 0.94 |
| **Mean \*** | **0.80** | **0.90** | **0.77** | **0.93** | **0.79** | **0.91** | **0.82** | **0.93** | **0.85** | **0.97** |

\* The final mean only with the algorithms selected for the ensemble. The algorithms that were selected for the ensemble are highlighted in bold. \*\* The MaxEnt algorithm was generated separately and was not part of the ensemble.

The suitability maps for all species created by MaxEnt are shown in Figure 1A–E. All models generated by MaxEnt had good performances, as indicated by their AUC and TSS values, with values above 0.9 and 0.7, respectively, except for *D. pagei*, which presented an AUC value of 0.84. Nevertheless, this still represents a useful predictive model (Table 2). The ensemble maps created by Biomod2 also presented excellent performance, based on the average AUC and TSS values, as shown in Figure 2. All species had AUC values greater than 0.9, with *D. pagei* having slightly lower values in some algorithms, but still with an excellent average of 0.9. The algorithms presented varied TSS values, but all had an average above 0.7.

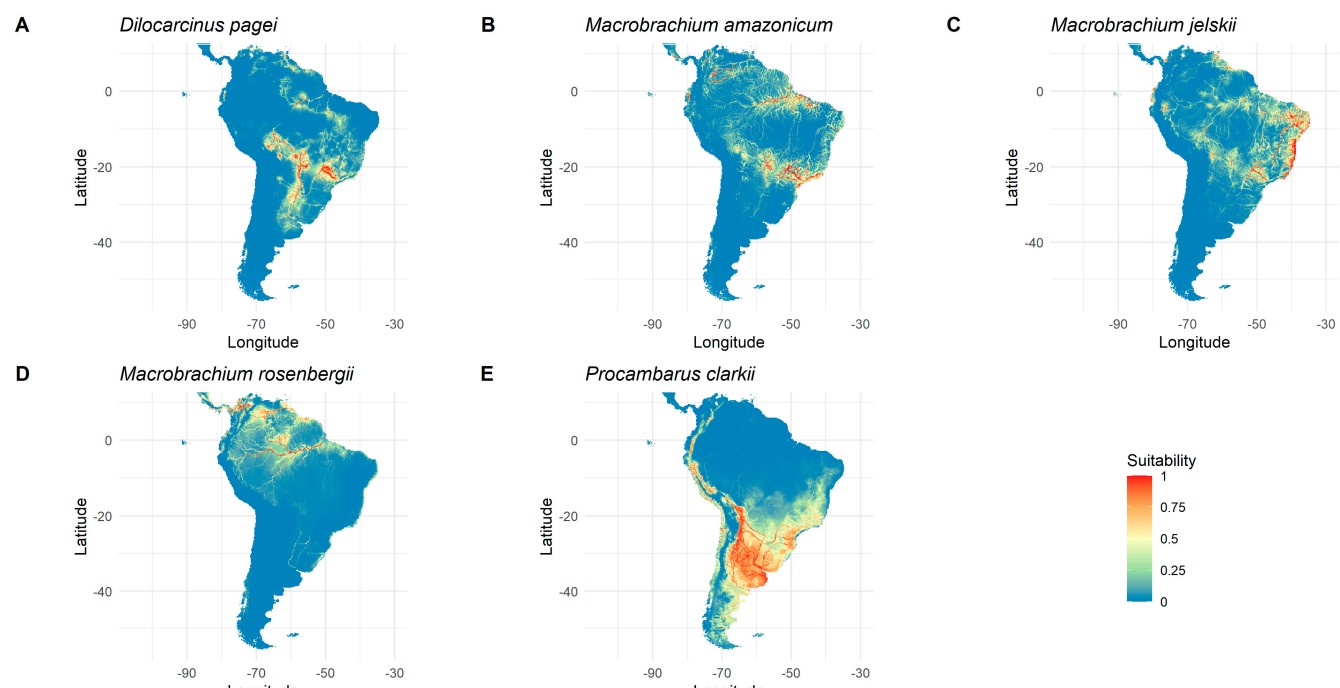

**Figure 1.** Suitability maps generated by the MaxEnt algorithm for each species: (**A**) *Dilocarcinus pagei*, (**B**) *Macrobrachium amazonicum*, (**C**) *Macrobrachium jelskii*, (**D**) *Macrobrachium rosenbergii*, (**E**) *Procambarus clarkii*. In red, suitability is greater than 75%.

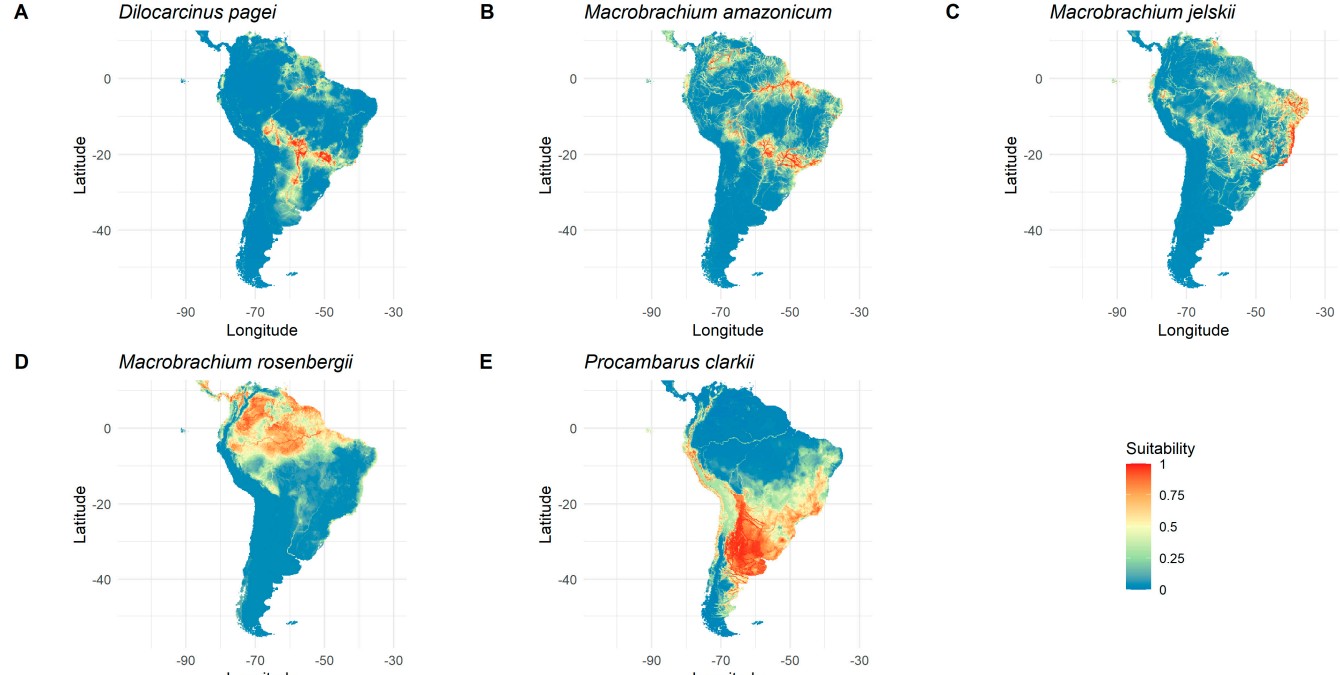

**Figure 2.** Suitability maps generated by the ensemble of the GLM, GBM, GAM, and RF algorithms for each species: (**A**) Dilocarcinus pagei, (**B**) Macrobrachium amazonicum, (**C**) Macrobrachium jelskii, (**D**) Macrobrachium rosenbergii, (**E**) Procambarus clarkii.

The ensemble result maps with niche suitability areas are shown in Figure 2A–E. *Dilocarcinus pagei* (Figure 2A), and *M. amazonicum* (Figure 2B) suitability maps were very similar, overlapping in almost the same hydrographic basins in the central region of South

America. The areas of greatest suitability are in the la Plata basin, in Brazilian territory and the middle portion of the Uruguay–Brazil basin and the Amazon River basin. The *M. jelskii* suitability map (Figure 2C) showed its greatest suitability areas close to the Brazilian coast that correspond to the river basins of East Brazil, the South Atlantic Coast, and Uruguay–Brazil, the South Atlantic Coast. Finally, the niche models for *M. rosenbergii* (Figure 2D) and *P. clarkii* (Figure 2E) showed excellent performance with the largest areas of suitability among all species. The areas of greatest suitability are in mainly in the Southern South America (la Plata River basin). More than 1.9 million km$^2$ of South American territory have a suitability greater than 0.75 for *P. clarkii*, while 953,640 Km$^2$ of South American territory is suitable for *M. rosenbergii* (Table 3).

**Table 3.** Suitability areas for each species modeled in square kilometers (Km$^2$) and the percentage relative to the territory of South America occupation. *P. clarkii* = *Procambarus clarkii*, *D. pagei* = *Dilocarcinus pagei*, *M. rosenbergii* = *Macrobrachium rosenbergii*, *M. jelskii* = *Macrobrachium jelskii*, *M. amazonicum* = *Macrobrachium amazonicum*.

| Species | Ensemble | | MaxEnt | |
| --- | --- | --- | --- | --- |
| | Suitability Area (km$^2$) | Suitability Area (%) | Suitability Area | Suitability Area (%) |
| *D. pagei* | 315,772.00 | 1.78 | 219,116.00 | 1.24 |
| *M. amazonicum* | 528,434.00 | 2.99 | 282,224.00 | 1.59 |
| *M. jelskii* | 313,870.00 | 1.77 | 280,939.00 | 1.59 |
| *M. rosenbergii* | 953,640.00 | 5.39 | 171,715.00 | 0.97 |
| *P. clarkii* | 1,962,112.00 | 11.09 | 846,132.00 | 4.78 |

Thus, we highlighted all the PAs and the main watersheds susceptible to invasion as priority areas for conservation, and this is shown in Supplementary Table S3. These areas were defined by overlapping the IUCN layers of PAs and hydrographic basins for South America, with the areas of suitability greater than 75% extracted from the maps of fitness of the five species modeled for this study.

## 4. Discussion

### 4.1. Suitability of Areas to Species Invasion

From the three that occur naturally in South America, *M. amazonicum* was the species that attracted the most attention in terms of its high invasion potential. This species is the most commercially exploited among the *Macrobrachium* species [56], and the results of the study highlight the urgent need for management strategies. Our results indicate that a large area (±3% of the entire territory of South America) is suitable for the occurrence of *M. amazonicum*. This area overlaps with several watersheds in the Southeast Region of Brazil and areas close to the Mato Grosso Pantanal, which also includes regions of Paraguay and Argentina [99], as well as areas of environmental protection.

The species that showed the greatest suitability area were *P. clarkii* and *M. rosenbergii*, with ~11% and 6% of the total South American territory being suitable for these species, respectively. *Procambarus clarkii* is a well-studied species that is ecologically and economically important. As a result, numerous modeling studies have been conducted to predict its invasive potential in both present and future scenarios, with global or regional projections. These studies have results similar to those presented in our study [5,48,100]. In addition, a study in Mexico demonstrated that *P. clarkii* can overlap habitats with native freshwater shrimp belonging to the genus *Macrobrachium*, resulting in significant impacts on the community of these species [101]. However, no affected organisms or impacts resulting from the presence of *P. clarkii* in South America have yet been described. Nevertheless, due to its great invasive potential, the presence of this species in aquatic ecosystems is a cause for concern [102,103].

For *M. rosenbergii*, little is known about the effects of introducing this species in South America, although there is evidence of invasions and impacts on native species on almost

all continents [51]. The Amazon region requires particular attention and containment strategies due to the Amazon Forest domain and nearly the entire Amazon River and Orinoco basin. Several native species occur in this region, including the native range of three of the five species in this study, as well as others with great ecological importance for the Amazon freshwater ecosystems [104,105]. The primary cause of the escape and establishment of *M. rosenbergii* in non-native environments is the lack of monitoring and maintenance of culture tanks [106], and all impacts discussed in the literature have only been described from other continents. This serves as a warning for a possible long-term biological invasion of *M. rosenbergii* in South America. Additionally, native species are at risk of competition for food and resources because *M. rosenbergii* can tolerate diverse environmental conditions [62].

The suitable areas modelled for *M. jelskii* were limited to regions close to the Brazilian coast, being a limiting factor for possible colonization within the continent, a natural impediment to the advancement of the species. However, the species has been found in interior regions, indicating that its ecological plasticity may allow for successful colonization in new areas. Anthropogenic actions may be the main cause of the introduction of this species outside the Amazon region, as is the case for *M. amazonicum* and *D. pagei*. Nonetheless, little is known about the biology and ecology of these species, and it is imprecise to calculate the impacts caused outside their natural distribution [105].

The presence of this species in coastal and estuarine regions within the Atlantic Forest domain, which includes several biological reserves, PAs, RPPN and Integral Protection Areas, raises concerns about its spread in South America. Although some authors consider *M. jelskii* an invasive species, the lack of studies demonstrating impacts makes this statement questionable [25,107,108].

*4.2. Niche Modeling*

The MaxEnt algorithm was used to generate models of the ecological niche for different species, with AUC values above 0.9 indicating good suitability areas, which were compared to models generated by other algorithms [83]. However, TSS values were lower for some species, with only *M. amazonicum* having TSS values above 0.8. The ensemble of the best models generated by the different algorithms was used to select the best models for each species, resulting in AUC means above 0.9 for all species. *Procambarus clarkii* had the highest TSS and AUC values due to a higher number of presence records. Sampling biases and noise were minimized with systematic reviews, which contributed to the species occurrence data by increasing the sample universe and reducing spatial bias [109]. Terrestrial variables, especially bioclimatic and topographic variables, were used to build the models. The lack of more accurate data for freshwater ecosystems makes the use of terrestrial variables advisable. However, for large extensions such as a country or continent, predictors with a high enough resolution and global scale are needed [76,110].

The use of MaxEnt and algorithms from the Biomod2 package brought relevant information to the present work because even using two different modeling tools, the results were similar, both in the suitability maps and in the model evaluation, with excellent results according to the literature. However, the differences observed in the suitability models generated by the MaxEnt algorithm and by the ensemble are due to the difference in the sensitivity and performance of the algorithms, which does not influence the quality of the model [111,112]. While the study provides new information on the potential niches of exotic and invasive species, further research is needed to better understand their distributions in Brazil. Future work should focus on collections and fieldwork in high-risk areas to verify whether they are already colonized, as well as projecting future results in conjunction with climate change data to gain more accurate information about species expansion.

Few studies used the overlapping of terrestrial and aquatic predictor variables simultaneously, especially when referring to crustaceans [113,114]. In this sense, by concomitantly associating terrestrial variables and aquatic variables as a single set of predictive variables, we managed to build a more robust model capable of mitigating the inherent deficits of each

database. This is the first work that modeled crustacean species associating the WorldClim terrestrial bioclimatic variables and EarthEnv aquatic variables.

Another corroboration of our results is that the training models, using only WorldClim data, presented results that were overfitted or without biological meaning and, when designed for the interest region, presented a low performance. Something similar occurred in [79] who also obtained a low predictive capacity. However, when associating the WorldClim variables with the IPCC predictive variables, which allowed for the contrasting of biases of the two variables, they exhibited greater environmental similarity between the training areas and testing, resulting in predictions with better performance. Some studies managed to predict the invasive capacity of certain species, including *P. clarkii*, by associating the Worldclim predictor variables with the aquatic variables available in the United States Great Lakes database [37,115]. Additionally, [116] showed that new SDM tools, including EarthEnv's unique freshwater variables, are estimating invasion probability with increasing accuracy, especially when combining local habitat data. However, it is almost unanimously agreed that the characterizations of freshwater conditions available in the databases are still primitive and with many limitations that need improvement to make the ENM even more robust. This condition still makes the aquatic species ENM dependent on terrestrial variables [76,117].

The contribution of precipitation variables to the model construction was significant for all species, as it aligns with the reproductive cycle and favorable environments for larval development. The rainy season, particularly from November to March, is correlated with the reproductive peak for some species such as *M. jelskii* [118,119]. High precipitation leads to a decrease in salinity, favoring embryonic development and larval growth, while dry seasons result in higher salinity and, in turn, favor reproduction in some species. The importance of precipitation variables in the models was further supported by studies showing the positive relationship between rain volume, nutrient input and decreasing the salinity [120–123]. In contrast, in the dry season, salinity is higher, favoring reproduction in these periods, resulting in a high production of eggs throughout the year in most species [14]. This factor corroborates the precipitation variables of the driest month (bio14 and hydro14) that were the most important for the *D. pagei* and *M. amazonicum* models, respectively.

The reproductive physiology of decapods is directly affected by air and water temperature, as they promote the growth of gonadal tissues [124–126]. In addition, the average diurnal interval (Hydro2), which analyzes the fluctuation of the maximum and minimum water temperature throughout the year, was important in the construction of the models of *M. jelskii* and *M. amazonicum*. Studies with *M. amazonicum* and other species of *Macrobrachium* reinforce that this temperature fluctuation can directly influence the number of individuals [12,127]. Finally, the average temperature of the warmest quarter (bio8) was important for the construction of the models for *M. rosenbergii* and *D. pagei*. Thus, the period of the year with the highest temperatures and the highest precipitation rates allows the decapod species to have more reproductive success, and these variables are important for their ecological niche definition.

## 5. Conclusions

Our results demonstrated that the Southeast Region of Brazil and the Pantanal are particularly vulnerable to invasion by freshwater decapod species. In addition, our study underscores the importance of the conservation and preservation of vulnerable habitats, particularly those located within Biological Reserves and Protected Areas, to prevent colonization by invasive species. However, we believe that our findings can inform efforts to protect aquatic and terrestrial biodiversity and guide the development of conservation and management measures to mitigate the risks of invasion and promote sustainable development.

**Supplementary Materials:** The following supporting information can be downloaded at: https://www.mdpi.com/article/10.3390/d15070841/s1, Figure S1: Flowchart of the PRISMA Matrix referring to the systematic literature review process; Figure S2: Occurrence maps of species; Table S1: Paper list resulting from the systematic review; Table S2: Predictor variables list, extracted from WorldClim and EarthEnv databases. Highlighted with an X are the variables selected through the Variance Inflation Factor (VIF) for each species; Table S3: Area of suitability greater than 75% of the species Dilocarcinus pagei, Macrobrachium amazonicum, Macrobrachium jelskii, Macrobrachium rosenbergii and Procambarus clarkii, which overlap the hydrographic basins and protected areas in South America.

**Author Contributions:** L.R.d.O. and S.J.C. conceived the idea; L.R.d.O., M.G., P.A. and S.J.C. designed the study; L.R.d.O., G.B. and S.J.C. organized the database; L.R.d.O., G.B., S.J.C. and X.M.C.O. performed the statistical analysis and models; L.R.d.O. wrote the first draft of the manuscript. All authors have read and agreed to the published version of the manuscript.

**Funding:** This study was financed in part by the Coordenação de Aperfeiçoamento de Pessoal de Nível Superior—Brasil (CAPES)—Finance Code 001.

**Institutional Review Board Statement:** Not applicable.

**Data Availability Statement:** All the results obtained are presented in Section 3 and the Supplementary Materials. Moreover, for more information or more details, the corresponding author can be contacted to obtain such information.

**Acknowledgments:** We are thankful to the CAPES institution for making this study possible, the reviewers who contributed timely to this article, and Sonia Sosa Quezada and Érica Cardoso de Lima for reviewing the English text. No fieldworks permits were required for this study.

**Conflicts of Interest:** The authors declare that the research was carried out without any commercial or financial relationships that could be construed as potential sources of conflicts of interest.

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
