# Peer review of "Non-Native Decapods in South America: Risk Assessment and Potential Impacts"

_diversity, doi:10.3390/d15070841_

Round 1

Reviewer 1 Report

Introducción

    -in the files 74 and 75 consider using more appropriate terminology, for example "transplant" (according to Darrigran, G., I. Agudo-Padrón, P. Baez, C. Belz, F. Cardoso, GA Collado, M. Correoso, MG Cuezzo, C. Damborenea, AA Fabres, MA Fernandez, S. R Gomes, DE Gutiérrez Gregoric, S. Letelier, C. Lodeiros, S. Ludwig, MC Mansur, S. Narciso, G. Pastorino, PE Penchaszadeh, AC Peralta, A. Rebolledo, A. . Rumi, RB Salvador, S. Santos, P. Spotorno, S. Carvalho Thiengo, T. Vidigal, A. Carranza -2022 online- Species movements within biogeographic regions: Exploring the distribution of transplanted mollusc species in South America . Biological invasions- BINV-D-21-00428R4 DOI: 10.1007/s10530-022-02942-z);

that is to say: define transplantation (transfer, translocation) as the movement of native species from one locality to another by human action, with successful establishment outside their known historical geographical range (in this case, within South America). Such species movements can be both accidental or deliberate (e.g., for conservation, trade, research). A species is considered to be established in the environment when there are populations with multi-generational reproductive success.

    - In files 112 and 113, "...the aims of this study..." based on what was said in the previous item, they are M. amazonicum, M. jelskii and D. pagei are transplant, and 2 non-native species in South America Procambarus clarkii (Girard 1852) and Dilocarcinus pagei (Stimpson 1861)

Materials and Methods 

Study area

    -In file 121 It is not La Plata, but the Plata (the Plata basin)

In files 121, 350, 357 You should not say "La" Plata, but "la" Plata, with a small l (when the Río de la Plata was named, no reference was made to the city of La Plata built 200 years later, but to la silver-in Spanish Silver=Plata-that's why it's "la Plata", Río de la Plata river)

Author Response

Response to Reviewer 1 Comments

Point 1:  -in the files 74 and 75 consider using more appropriate terminology, for example "transplant" (according to Darrigran, G., I. Agudo-Padrón, P. Baez, C. Belz, F. Cardoso, GA Collado, M. Correoso, MG Cuezzo, C. Damborenea, AA Fabres, MA Fernandez, S. R Gomes, DE Gutiérrez Gregoric, S. Letelier, C. Lodeiros, S. Ludwig, MC Mansur, S. Narciso, G. Pastorino, PE Penchaszadeh, AC Peralta, A. Rebolledo, A. . Rumi, RB Salvador, S. Santos, P. Spotorno, S. Carvalho Thiengo, T. Vidigal, A. Carranza -2022 online- Species movements within biogeographic regions: Exploring the distribution of transplanted mollusc species in South. Biological invasions- BINV-D-21-00428R4 DOI: 10.1007/s10530-022-02942-z);

that is to say: define transplantation (transfer, translocation) as the movement of native species from one locality to another by human action, with successful establishment outside their known historical geographical range (in this case, within South America). Such species movements can be both accidental or deliberate (e.g., for conservation, trade, research). A species is considered to be established in the environment when there are populations with multi-generational reproductive success.

Response: We appreciate your suggestion. We have understood that the ideal term to define the introduced species is "transplantation" and we have included the suggested changes, as well as in other points of the manuscript. We have added the references as suggested (lines 79, 80, 125, 126 and 148)

Although reviewer 3 suggested shortening the paragraph, we decided to retain it to incorporate the term 'transplant' as it holds relevance to our study. We acknowledge that the explanation of this term and the classification of species are essential for proper understanding.

Point 2: - In files 112 and 113, "...the aims of this study..." based on what was said in the previous item, they are M. amazonicum, M. jelskii and D. pagei are transplant, and 2 non-native species in South America Procambarus clarkii (Girard 1852) and Dilocarcinus pagei (Stimpson 1861)

Response: We have changed the original sentence as suggested (line 116-117)

Point 3:

Materials and Methods

Study area

 -In file 121 It is not La Plata, but the Plata (the Plata basin)

In files 121, 350, 357 You should not say "La" Plata, but "la" Plata, with a small l (when the Río de la Plata was named, no reference was made to the city of La Plata built 200 years later, but to la silver-in Spanish Silver=Plata-that's why it's "la Plata", Río de la Plata river)

Response: We appreciate your suggestion. We have changed the word as suggested (line 124, 353, 360)

Reviewer 2 Report

Although generally well-written, there are some grammatical issues in places and also a few areas where the sentences appear incomplete.

Line 28: suitable habitat areas

Lines 29-30: [something is missing in this sentence]

Line 32: indicate a

Line 35: identifying areas susceptible to

Line 39: distributions have

Line 43: the ecosystem, while also

Line 45: due to climate change and degredation

Line 50: Freshwater macro- and

Line 53: Decapods that are adapted

Lines 53-53: ecosystems can tolerate different… variations which gives them

Line 58: cycles

Line 61: and represents approximately

Line 62: biodiverse regions

Line 72: potential. Procambarus clarkii and M. rosenbergii are native to North

Line 74: to the Amazon

Line 75: regions of South America

Lines 79-80: [something is missing in this sentence]

Line 80: Procambarus clarkii is [don’t abbreviate the genus when it starts a sentence]

Line 87: pagei, are native

Line 88: Macrobrachium jelskii

Line 90: While the introduction of D. pagei in other regions is believed to have been accidental,

Line 100: invasions [37,38] by

Line 112: were to: 1) model the ecological niches [you can’t say they all have the SAME niche]

Line 114: habitat areas

Line 117: territory is located at

Line 123: basin; (3) the Guiana

Line 124: ; and (4) the Central

Line 133: [non-native? Is that really what you mean?]

Line 137: as is also true for M. jelskii

Line 145: notable within the genus… inland areas and is not

Line 162: south-central United States

Line 167: review and to compile

Line 193: suitable niches

Line 242: species to reduce

Line 311: as one of the three

Line 342: (Table 2)

Line 343: the average AUC

Line 348: (Figure 2A)

Line 355: (Figure 2E)

Line 374: and this is shown in Supplementary Table S3. These areas were

Line 375: defined by overlapping the IUCN

Lines 380-381: The species, of the three that occur naturally in South America, that attracted the most attention in terms of high potential for invasion was M. amazonicum.

Line 393: These studies have results similar to those

Line 394: Mexico demonstrated that

Line 396: of those species

Line 401: although there is evidence

Line 405: species in this study

Line 406: [you say “as well as others” so give example(s) of such species in Amazon FW ecosystems]

Line 408: have only been described from other

Line 411: resources because M. rosenbergii

Lines 413-414: [something is missing in this sentence]

Line 416: actions may

Line 419: outside their natural

Line 424: lack of studies

Line 432: Procambarus clarkii had

Line 446: potential niches

Line 447: their distributions

Line 455: each database. This is the first work

Line 457: Another corroboration of our results

Line 464: of certain species

Line 466: Additionally, [114] showed

Line 469: unanimously agrees that

Line 476: species such as M. jelskii

Line 478: favor reproduction

Line 480: descreasing the

Line 491: of Macrobrachium

Line 493: models for M. rosenbergii

References 24, 38 and 97 (at least) need italics on taxon names. References should be proofread carefully.

See above comments.

Author Response

Response to Reviewer 2 Comments

Line 28: evaluate remaining suitable habitat areas for the occurrence of those species in South America.

Response: We have changed the word “habitats” as suggested (page 1, line 26 and 27).

Lines 29-30 [something is missing in this sentence]

Response: Something was certainly missing from the abstract. We appreciate your suggestion and we have changed the original sentence as recommended (page 1, lines 30-31)

Line 32: indicate a

Response: We have changed the original sentence as suggested (page 1, lines 3 and 34)

Line 35: identifying areas susceptible to

Response: We have changed the original sentence as suggested (page 1, line 37)

Line 39: distributions have

Response: We have changed the word as suggested (page 1, line 41)

Line 43: the ecosystem, while also

Response: We have included the word “while” as suggested (page 1, line 45)

Line 45: due to climate change and degradation

Response: We have changed the original sentence as suggested (page 1, line 47)

Line 50: Freshwater macro- and

Response: We have changed the original sentence as suggested (page 2, line 52)

Line 53: Decapods that are adapted to

Response: We have changed the original sentence as suggested (page 2, line 55)

Lines 53-54: ecosystems can tolerate different… variations which gives them

Response: We have removed the commas from the original sentence as suggested (page 2, line 56)

Line 58: cycles

Response: We have changed the word as suggested (page 2, line 60)

Line 61: and represents approximately

Response: We have changed the original sentence as suggested (page 2, lines 61 and 62)

Line 62: biodiverse regions

Response: We have changed the original word as suggested (page 2, line 63)

Line 72: potential. Procambarus clarkii and M. rosenbergii are native to North America.

Response: We have changed the original sentence to make it clear (page 2; lines 72 and 73).

Line 74: to the Amazon

Response: We have included the word as suggested (page 2; line 76).

Line 75: regions of South America

Response: We have included the word as suggested (page 2; lines 79 and 80).

Lines 79-80: [something is missing in this sentence]

Response: We appreciate your suggestion. We have rephrased the original sentence as suggested (page 2 lines 78-80).

Line 80: Procambarus clarkii is [don’t abbreviate the genus when it starts a sentence]

Response: We have included the full genus name as suggested (page 2, line 86)

Line 87:  pagei, are native

Response: We have included the comma in the sentence as suggested (page 2, line 94)

Line 88:  Macrobrachium jelskii

Response: We have included the full genus name as suggested (page 2, line 95)

Line 90: While the introduction of D. pagei in other regions is believed to have been accidental,

Response: We have changed the original sentence to make it clear (page 2; lines 97-99).

Line 100: invasions [37,38] by

Response: We have included the comma in the sentence as suggested (page 3, line 110)

Line 112: were to: 1) model the ecological niches [you can’t say they all have the SAME niche]

Response: We appreciate your suggestion. We have changed the original word as suggested (page 3, line 125).

Line 114: habitat areas

Response: We have changed the original word as suggested (page 3, line 127)

Line 117: territory is located at

Response: We have changed the original word as suggested (page 3, line 131).

Line 123: basin; (3) the Guiana

Response: We have corrected the punctuation in the sentence (page 3, line 137).

Line 124: ; and (4) the Central

Response: We have corrected the punctuation in the sentence (page 3, line 138).

Line 133: [non-native? Is that really what you mean?]

Response: Response: We appreciate your suggestion. We have understood that the ideal term to define the introduced species is "transplantation" and we have included the suggested changes, as well as in other points of the manuscript. We have added the references as suggested (lines 79, 80, 125, 126 and 148)

Line 137: as is also true for M. jelskii"

Response: We have rephrased the original sentence as suggested (page 3, line 151)

Line 145: notable within the genus… inland areas and is not

Response: We appreciate your suggestion. We have rephrased the original sentence as suggested (page 3, line 159).

Line 162: south-central United States

Response: We have changed the original sentence as suggested (page 4, line 177)

Line 167: review and to compile

Response: We have changed the original sentence as suggested (page 4, line 182)

Line 193: suitable niches

Response: We have included the word as suggested (page 4; line 208).

Line 242: species to reduce

Response: We have included the word as suggested (page 5; line 259).

Line 311: as one of the three

Response: We have changed the original sentence to make it clear (page 7; lines 328).

Line 342: (Table 2)

Line 348: (Figure 2A)

Line 355: (Figure 2E)

Response: We have included the capital letters on Figures and Tables as suggested (page 8; lines 359, 365 and 372).

Line 343: on the average AUC

Response: We have changed the word as suggested (page 8; line 360).

Line 374: and is shown on Supplementary Table S3. This areas were defined

and this is shown in Supplementary Table S3. These areas were

Response: We have changed the original sentence to make it clear (page 10; line 391-393).

Line 375: defined by overlapping the IUCN

Response: We appreciate your suggestion. We have changed the original sentence to make it clear (page 10; lines 392 and 393).

Lines 380-381: The species, of the three that occur naturally in South America, that attracted the most attention in terms of high potential for invasion was M. amazonicum.

Response: We have changed the original sentence to make it clear (page 9; lines 398-401).

Line 393: These studies have results similar to those

Response: We have changed the original sentence to make it clear (page 10; line 414).

Line 394: Mexico demonstrated that

Response: We have changed the word as suggested (page 10; line 415).

Line 396: of those species

Response: We have changed the word as suggested (page 10; line 417).

Line 401: although there is evidence

Response: We have changed the original sentence to make it clear (page 10; line 422).

Line 405: species in this study

Response: We have removed the redundancy from the sentence (page 10; line 426).

Line 406: [you say “as well as others” so give example(s) of such species in Amazon FW ecosystems]

Response: We appreciate your suggestion. We have added new references about these species in Amazon FW ecosystems (page 10; line 427).

Line 408: have only been described from other

Response: We have changed the original sentence to make it clear (page 10; lines 429-430).

Line 411: resources because M. rosenbergii

Response: We have changed the word as suggested (page 11; line 432).

Lines 413-414: [something is missing in this sentence]

Response: We appreciate your suggestion. We have rephrased the original sentence as suggested (page 11, lines 434-438).

Line 416: actions may

Response: We have removed the comma in the original sentence as suggested (page 11; line 440)

Line 419: outside their natural

Response: We have changed the word as suggested (page 11; line 444).

Line 424: lack of studies

Response: We have added the word as suggested (page 11; line 448).

Line 432:  Procambarus clarkii had

Response: We have included the full genus name as suggested (page 11, lines 456 and 457)

Line 446: potential niches

Response: We have changed the word as suggested (page 11; line 472).

Line 447: their distributions

Response: We have changed the word as suggested (page 11; line 473).

Line 455: each database. This is the first work

Response: We appreciate your suggestion. We have rephrased the original sentence as suggested (page 11, line 481).

Line 457: Another corroboration of our results

Response: We appreciate your suggestion. We have rephrased the original sentence as suggested (page 12, line 484).

Line 464: of certain species

Response: We have changed the word as suggested (page 12; line 491).

Line 466: Additionally, [114] showed

Response: We have changed the word as suggested (page 12; line 493).

Line 469: unanimously agrees that

Response: We have rephrased the original sentence as suggested (page 12, line 496).

Line 476: species such as M. jelskii

Response: We have changed the word as suggested (page 12; line 504).

Line 478: favor reproduction

Response: We have changed the word as suggested (page 12; line 506).

Line 480: descreasing the

Response: We have changed the word as suggested (page 12; line 509).

Line 491: of Macrobrachium

Response: We have removed the word as suggested (page 12; line 519).

Line 493: models for M. rosenbergii

Response: We have removed the word as suggested (page 12; line 521).

References 24, 38 and 97 (at least) need italics on taxon names. References should be proofread carefully.

Response: We have revised all references and taxon names have been added in italics.

Reviewer 3 Report

 The paper is very interesting and brings robust analyses. I believe that it will lead other researchers to apply this methodology in their research! The English is clear, and the figures are of high quality.

I would like to mention that the authors, avoid redundancy! The paper per se deals with invasive species, thus it is not necessary to reinforce in several parts the negative effects of them, it is clear! Also, try to follow one way of writing, of the first or third person. Please, check this along with the manuscript.

Author Response

Response to Reviewer 3 Comments

Point 1: The paper is very interesting and brings robust analyses. I believe that it will lead other researchers to apply this methodology in their research! The English is clear, and the figures are of high quality.

I would like to mention that the authors, avoid redundancy! The paper per se deals with invasive species, thus it is not necessary to reinforce in several parts the negative effects of them, it is clear! Also, try to follow one way of writing, of the first or third person. Please, check this along with the manuscript.

Response: We thank the reviewer for the valuable comments and suggestions. We revised the manuscript considering the points raised in the pdf version. All changes suggested were made accordingly and are highlighted in the word track change version of the manuscript. Furthermore, although the reviewer suggested shortening the paragraph (line 122-128), we decided to retain it to incorporate the term 'transplant' as it holds relevance to our study. We acknowledge that the explanation of this term and the classification of species are essential for proper understanding.
